# Chemical Composition, Antioxidant and Anti-Inflammatory Activity of Shiitake Mushrooms (*Lentinus edodes*)

**DOI:** 10.3390/jof10080552

**Published:** 2024-08-05

**Authors:** Xiaoming Xu, Chong Yu, Zhenyang Liu, Xiaohang Cui, Xiaohe Guo, Haifeng Wang

**Affiliations:** 1Key Laboratory of Structure-Based Drug Design & Discovery of Ministry of Education, School of Traditional Chinese Materia Medica, Shenyang Pharmaceutical University, Shenyang 110016, China; 18040077532@163.com (X.X.); yuchong0408@163.com (C.Y.); cuixiaohang1860@163.com (X.C.); xiao154953@163.com (X.G.); 2Guangxi Key Laboratory of Marine Natural Products and Combinatorial Biosynthesis Chemistry, Guangxi Academy of Sciences, Nanning 530007, China; 3Shenyang Baide Biotechnology Co., Ltd, Shenyang 110016, China; 13898161725@163.com; 4Key Laboratory of Innovative Traditional Chinese Medicine for Major Chronic Diseases of Liaoning Province, School of Traditional Chinese Materia Medica, Shenyang Pharmaceutical University, Shenyang 110016, China

**Keywords:** shiitake mushroom (*Lentinus edodes*), edible fungi, mycelium and fruiting body, bioactive compounds, anti-inflammatory activity, antioxidant activity

## Abstract

Shiitake mushrooms (*Lentinus edodes*) are renowned as the “King of mountain treasures” in China due to their abundant nutritional and health-enhancing properties. Intensive chemical investigations of the fruiting bodies and mycelium of Shiitake mushrooms (*Lentinus edodes*) afforded five new compounds (**1–5**), named lentinmacrocycles A-C and lentincoumarins A-B, along with fifteen known compounds (**6–20**). Their structures and absolute configurations were elucidated by extensive spectroscopic analysis, including one-and two-dimensional (1D and 2D) NMR spectroscopy, circular dichroism (CD), and high-resolution electrospray ionization mass spectrometry (HR-ESI-MS). The anti-inflammatory activity test showed that lentincoumarins A (**4**), (3*S*)-7-hydroxymellein (**9**), (3*R*)-6-hydroxymellein (**11**) and succinic acid (**18**) exhibited strong NO inhibitory effects (IC_50_ < 35 μM), and that (3*S*)-5-hydroxymellein (**10**) and (3*R*)-6-hydroxymellein (**11**) exhibited potent TNF-α inhibitory effects (IC_50_ < 80 μM) and were more potent than the positive control, Indomethacin (IC_50_ = 88.5 ± 2.1 μM). The antioxidant activity test showed that (3*R*)-6-hydroxymellein (**11**) had better DPPH radical scavenging activity (IC_50_ = 25.2 ± 0.5 μM).

## 1. Introduction

Most edible fungi are rich in bioactive substances, which not only have a high nutritional value but also have medicinal properties; therefore, they have attracted much attention [1]. At present, research on edible fungi has gradually shifted from traditional nutritional foods to health foods. Researchers have attached great importance to the development of edible fungi with good efficacy [2,3,4]. As a representative large edible fungus, shiitake mushrooms (*Lentinula edodes*) have a long history of consumption and are widely popular worldwide because of their attractive taste [5,6]. Bioactive compounds such as polysaccharides, purines, proteins (amino acids), fatty acids, polyphenols, and sterols [7,8] in shiitake mushrooms have been shown to have high nutritional value and enhance human health by promoting anti-inflammatory, antioxidant, antitumor, antiviral, antibacterial, and immunostimulatory effects [9,10,11,12,13].

At present, reports on the active substances in shiitake mushrooms have mostly focused on polysaccharides [14,15]. Other types of bioactive compounds in shiitake mushrooms have often been overlooked. The growth of *Lentinus edodes* can be divided into two stages: the vegetative phase (mycelium or mycelial growth) and the reproductive phase (fruiting bodies). After the scattered spores have invaded the substrate, the hyphae, which are only visible under a microscope, continually grow and branch to form mycelia, and the fruiting body grows out of the subterranean mycelium through a process called fructification. The fruiting bodies (at the bottom of the cap) are sporulation structures that are edible parts of edible fungi [16,17]. In recent years, research on medicinal fungi has expanded from the initial fruiting bodies to liquid fermentation [18], the number of functional secondary metabolites isolated has increased year by year, and their structures have become increasingly diversified [19].

In our study, we focused on the non-polysaccharide components of the fruiting bodies and mycelia of shiitake mushrooms. Ten compounds were obtained from the fruiting bodies and mycelia, and their anti-inflammatory and antioxidant properties were studied.

## 2. Materials and Methods

### 2.1. Instrument and Materials

UV data were collected on a Shimadzu UV-1700 spectrometer (Shimadzu, Kyoto, Japan). The IR spectra were measured on a Bruker IFS-55 (Bruker, Karlsruhe, Germany). The optical rotation values were obtained using a Perkinelmer 241 MC polarimeter (Perkinelmer, Waltham, MA, USA). The CD spectrum was measured by a Bio-Logic Science Instruments spectrometer (Bio-Logic Science Instruments, Seyssinet-Pariset, France). HRESIMS data were collected using a Thermo Scientific Orbitrap Fusion Lumos Tribrid mass spectrometer (Thermo Scientific, Waltham, MA, USA). The NMR spectra were recorded using a Bruker AVANCEIII-600 and AVANCEIII-400 NMR spectrometer (Bruker Corporation, Karlsruhe, Germany). HPLC separation was carried out on a semi-preparative YMC-pack ODS-A column (250 × 10 mm, YMC-pack, Kyoto, Japan) equipped with a Shimadzu LC-8A detector (Shimadzu, Kyoto, Japan). Chromatographic silica gel (100–200 and 200–300 mesh, Yuwang Reagent Co., Ltd., Qingdao, China) and Sephadex LH-20 (18–110 lm; Pharmacia, Piscataway, NJ, USA), were used for column chromatography. The thermostatic shaker used for cultivating fungi uses an ZQPL-200 shaker (Tianjin Laiboteri Instrument Equipment Co., Ltd., Tianjin, China)

### 2.2. Reagents and Chemicals

The HPLC-grade solvents, such as methanol and acetonitrile (99.9%), and the analytical reagent solvents, such as petroleum ether, dichloromethane, ethyl acetate, and ethanol (99.5%), were purchased from Shandong Yuwang Reagent Co., Ltd. (Qingdao, China). The MTT used to detect the cells was obtained from Sigma Aldridge, St. Louis, MO, USA. The DPPH reagent used to detect the antioxidant activity of cells was derived from Qingbei Reagent Co., Ltd. Jinan 250100, China.

### 2.3. Fungal Material

Secondary strains of *Lentinus edodes* (with 78% wood chips, 20% bran, 1% sucrose, and 1% gypsum) were collected from the Edible Fungi Research Base of Shenyang Agricultural University and deposited at Shenyang Pharmaceutical University. After the colony grew, the strain was purified to a single colony, namely *Lentinus edodes* strain LE-M. Fresh shiitake mushrooms were obtained from Shenyang Farmers’ Market. Professor Pei of Shenyang Pharmaceutical University identified that the quality of this batch of shiitake mushrooms was excellent.

### 2.4. Large-Scale Fermentation of Strain LE-M

Preparation of the seed liquid: The purified mycelia of LE-M were inoculated in a 250 mL conical flask containing 150 mL of fungal No. 4 medium (1 L: 2% mannitol; 2% glucose; 0.5% yeast extract; 1% peptone; 0.05% KH_2_PO_4_; 0.03% MgSO_4_·7H_2_O; 0.1% corn steep liquor; water) and shaken at 25 °C for 10 days at 170 rpm. Afterward, the seed liquid (5 mL) was transferred to rice medium (100 g of rice and 100 mL of water, 220 bottles) and fermented at 25 °C statically for 45 days [20,21,22].

### 2.5. The Isolation of Chemical Components

*Lentinus edodes* mycelium rice fermentation was extracted with EtOAc 3 times at room temperature to obtain an EtOAc layer extract (72.5 g). The extract of the organic layer was subjected to column chromatography over silica gel (200–300 mesh, CH_2_Cl_2_-MeOH, step gradient elution 1:0, 50:1, 25:1, 10:1, 5:1, 2:1, 1:1 and 0:1) to obtain seven fractions (Fr. 1–Fr. 7). Fr. 1 (6.0 g) was separated on a silica gel column (200–300 mesh, PE-acetone, step gradient elution 10:1, 4:1, 2:1, and 1:1) to obtain six fractions (Fr. 11–Fr. 16). Then, Fr. 14 (2.6 g) was purified by Sephadex LH-20 (CH_3_OH) to obtain compound **5** (12.5 mg). Fr. 15 (1.9 g) was purified by pre-HPLC (MeOH/H_2_O = 35:65, 2.0 mL/min) to obtain compound **4** (4.3 mg, tR = 23.0 min).

The fruiting body of shiitake mushrooms was extracted with 95% ethanol 3 times to obtain a 25 L concentrate. The concentrate was extracted by EtOAc to yield a crude extract after evaporation under vacuum. The crude extract (19.6 g) of the organic layer was subjected to column chromatography over silica gel (200–300 mesh, CH_2_Cl_2_-MeOH, step gradient elution 1:0, 50:1, 25:1, 10:1, 5:1, 2:1, 1:1 and 0:1) to obtain eight subfractions (Fr. G1–Fr. G8). Fr. G5–2 (3.8 g) was purified by pre-HPLC (MeOH/H_2_O = 60:40, 2.0 mL/min) to obtain compound **1** (24.1 mg, tR = 15.6 min), compound **2** (55.6 mg, tR = 17.8 min), and compound **3** (1.6 mg, tR = 19.7 min).

(3*S*,5*S*)-5-hydroxy-2,3,4,5-tetrahydrobenzo[b]oxepine-3,7-diyl (**1**): Yellow oil (MeOH); HR-ESI-MS: *m/z* 247.0941 [M + Na]^+^ (calcd for C_12_H_16_O_4_Na, 247.0930); IR (KBr): 3396, 1634 cm^−1^; [*α*${]}_{D}^{25}$-6.0; UV: 254 nm; the ^1^H NMR and ^13^C NMR measured using the reagent DMSO-*d*_6_ are shown in Table 1. The supplementary spectra of compound **1** are shown in Appendix A.

(*R*)-3-Hydroxy-3,7-bis(hydroxymethyl)-3,4-dihydrobenzo[b]oxepin-5(2*H*)-one (**2**): Yellow oil (MeOH); [*α*${]}_{D}^{25}$-13.0 (c 0.01, MeOH). HR-ESI-MS: *m*/*z* 261.0747 [M + Na]^+^ (calcd for C_12_H_14_O_5_Na, 261.0750); IR (KBr): 2929, 2878, 1676 cm^−1^; UV: 254 nm; the ^1^H NMR and ^13^C NMR measured using the reagent DMSO-*d*_6_ are shown in Table 1. The supplementary spectra of compound **2** are shown in Appendix A.

(3-Hydroxy-3-(hydroxymethyl)-5-oxo-2,3,4,5-tetrahydrobenzo[b]oxepin-7-yl) methyl-2’-oxopyrrolidine-5’-carboxylate (**3**): Yellow oil (MeOH); [*α*${]}_{D}^{25}$+ 6.0 (c 0.01, MeOH). HR-ESI-MS gave *m*/*z* 372.1043 [M + Na]^+^ (calcd for C_17_H_19_O_7_NNa, 372.1040); IR (KBr): 2926, 1738, 1679 cm^−1^; UV: 254 nm; the ^1^H NMR and ^13^C NMR measured using the reagent DMSO-*d*_6_ are shown in Table 1. The supplementary spectra of compound **3** are shown in Appendix A.

(3*S*,8*S*)-3-(1-Hydroxy-but-3-ynyl)-3*H*-isobenzofuran-1-one (**4**): Yellow oil (MeOH); HR-ESI-MS *m*/*z* 225.0648 [M + Na]^+^ (calcd for C_12_H_10_O_3_Na, 225.0630); IR (KBr): 3429, 1671, 1472, 1384, 1291, 1202, 1053, and 1017 cm^−1^; UV peak at 347 nm and 205 nm; the ^1^H NMR and ^13^C NMR measured using reagent DMSO-*d*_6_ are shown in Table 2. The supplementary spectra of compound **4** are shown in Appendix A.

(3*S*,8*R*)-3-(1-Hydroxy-but-3-ynyl)-3*H*-isobenzofuran-1-one (**5**): Yellow oil (MeOH); HR-ESI-MS *m*/*z* 225.0648 [M + Na]^+^ (calcd for C_12_H_10_O_3_Na, 225.0630); IR (KBr): 3429, 1671, 1472, 1384, 1291, 1202, 1053, 1017 cm^−1^; UV: 347 nm, 205 nm; the ^1^H NMR and ^13^C NMR measured using the reagent DMSO-*d*_6_ are shown in Table 2. The supplementary spectra of compound **5** are shown in Appendix A.

### 2.6. Antioxidant Activity Assays

A 150 μMol/L DPPH solution was diluted with MeOH, and the final concentration reached an absorbance of 1.0 at a wavelength of 517 nm. Then, 3 mL of DPPH solution diluted with 1 mL of sample was mixed on 96-well plates. The absorbance was measured using a Varoskan flash multimode reader. DPPH absorbance inhibition rate (%) = (Acontrol − Atest) × 100/Acontrol [23].

### 2.7. Measurement of Nitrite Production and TNF-α

After 48 h of culture, the RAW264.7 cells produced NO in the medium, and 100 mL of supernatant was collected and reacted with 100 mL of Griess reagent. Indomethacin served as a positive control. The amount of NO produced at 570 nm was measured. The TNF-*α* levels were measured using an ELISA kit. The TNF-*α* detection method used was based on the literature [24,25].

### 2.8. MTT Assay

The activity of RAW264.7 cells was tested by MTT (Sigma Aldrich, Missouri, USA) colorimetry after 48 h of incubation, after which different concentrations of compounds were added. Then, 10 mL of 5 mg/mL MTT was added to the 96-well plates. The cells were incubated at 37 °C for 4 h, after which the medium was removed. The formazan produced by the reduction dye from live cells was lysed with 0.04 mol/L HCl isopropanol solutions. Finally, the optical density was measured. When the optical density of the experimental compound group was less than 80% of that of the control group, the compounds in the experimental group were considered to be cytotoxic.

## 3. Results

### 3.1. Structure Elucidation of Components

The non-polysaccharide components of the fruiting bodies and mycelia of *Lentinus edodes* were studied. Ten compounds were isolated, respectively (Figure 1). Among them, there were 5 new compounds and 15 known compounds (The supplementary spectra of known compounds are shown in Appendix A) [26,27,28,29,30,31]. Next, the structure of the new compound was elucidated.

Compound **1** was a yellow oil (MeOH), HR-ESI-MS data (*m*/*z* 247.0941 [M + Na]^+^), with a molecular formula of C_12_H_16_O_4_, with five degrees of unsaturation. The ^1^H NMR spectrum (Table 1, Appendix A) displayed three hydroxy hydrogen signals at *δ*_H_ 5.40 (1H, s), 5.10 (1H, s), and 4.62 (1H, s); six oxymethylene hydrogen signals at *δ*_H_ 4.43 (2H, s, H-11), 4.31 (1H, dd, *J* = 11.0, 3.8 Hz, H-2), 3.31 (1H, m, H-10), 3.23 (1H, m, H-10), and 3.17 (1H, d, *J* = 11.0 Hz, H-2); two methylene hydrogen signals at *δ*_H_ 2.02 (1H, d, *J* = 12.0 Hz, H-4) and 1.28 (1H, q, *J* = 12.0 Hz, H-4); one oxymethine at *δ*_H_ 4.72 (1H, d, *J* = 11.0 Hz, H-5); and one methine at *δ*_H_ 2.15 (1H, m, H-3). In addition, the hydrogen signals at *δ*_H_ 7.48 (1H, s, H-6), 7.06 (1H, dd, *J* = 8.5, 2.0 Hz, H-8), and 6.85 (1H, d, *J* = 8.1 Hz, H-9) suggested that compound **1** contains a set of ABX diphenyl ring proton signals. ^13^C-NMR (150 MHz, DMSO-*d*_6_) combined with the HSQC spectrum revealed aromatic carbons at *δ*_C_ 156.5 (C-9a), 138.7 (C-5a), 137.6 (C-7), 125.9 (C-8), 124.7 (C-6), and 120.5 (C-9); four oxygen-aliphatic signals at *δ*_C_ 75.0 (C-2), 68.7 (C-5), 63.4 (C-11), and 62.8 (C-10); and two aliphatic signals at *δ*_C_ 42.5 (C-3) and 39.6 (C-4).

The HMBC (Figure 2 and Appendix A) correlations from H-6 to C-5/C-5a/C-11, from H-8 to C-6/C-11, and from H-9 to C-7/C-9a indicated a benzene ring. C-7 was linked to C-11, and C-5 was linked to C-5a. The correlations from H-10 to C-2/C-3/C-4, from H-4 to C-3/C-5, and from H-2 to C-10/C-9a indicated a seven-membered ring structure, and C-3 was connected to C-10. Consequently, compound **1** was elucidated.

The relative structure of compound **1** could be determined by analyzing its NOESY correlation spectrum (Appendix A). H-5 was correlated with H-3 in the NOESY spectrum; therefore, H-3 and H-5 were on the same side. After complexation with a metal rhodium salt, the absolute configuration of C-5 was elucidated by the CD spectrum (Figure 3). According to the bulkiness rule, there was a positive Cotton effect at 350 nm [32,33]; hence, C-5 was elucidated as *S*. Combined with the relative configuration; C-3 was *S*-type.

Thus, compound **1** was elucidated, and the systematic name was (3*S*,5*S*)-5-hydroxy-2,3,4,5-tetrahydrobenzo[b]oxepine-3,7-diyl.

Compound **2** was a yellow oil (MeOH), HR-ESI-MS (*m*/*z* 261.0747 [M + Na]^+^), with a molecular formula of C_12_H_14_O_5,_ with six degrees of unsaturation. ^1^H and ^13^C NMR (Table 1; Appendix A) suggested that compounds **1** and **2** have very similar structures, with the only significant difference being the substituents on the seven-membered ring. *δ*_C_ 196.6 (C-5) was a ketocarbonyl group, and *δ*_C_ 76.7 (C-3) was a quaternary carbon with oxygen. The HMBC (Figure 2) correlations from H-6 to C-5, from H-4 to C-2/C-3/C-5/C-5a/C-10, from H-10 to C-2/C-3, and from H-2 to C-3/C-9a confirmed the planar structure of compound **2**. The absolute configuration of C-3 was determined by the CD spectrum (Figure 3 and Appendix A) after complexation with a metal rhodium salt [32,33]. A negative Cotton effect at 350 nm elucidated the absolute configuration of C-3 as *R*.

Thus, the structure of compound **2** was determined, and the systematic name was (*R*)-3-hydroxy-3,7-bis(hydroxymethyl)-3,4-dihydrobenzo[b]oxepin-5(2*H*)-one.

Compound **3** was a white powder, HR-ESI-MS (*m*/*z* 372.1043 [M + Na]^+^), with a molecular formula of C_17_H_19_O_7_N and eight degrees of unsaturation. The ^1^H NMR data suggested (Table 1; Appendix A) that compounds **2** and **3** have very similar structures, with the only significant difference being the substituent on the benzene ring. The HMBC (Figure 2 and Appendix A) correlations from N-H to C-3′/C-4′/C-5′, from H-3′ to C-2′/C-5′, and from H-11 to C-6′/C-7/C-8 and the ^1^H-^1^H COSY correlations of H-3′ with C-4a’/C-4b’ and H-4a’ with C-5′/C-4b’ established a five-membered nitrogen heterocyclic ring linked to C-11. The planar structure of compound **3** was elucidated, and the systematic name was (3-hydroxy-3-(hydroxymethyl)-5-oxo-2,3,4,5-tetrahydrobenzo[b]oxepin-7-yl) methyl-2’-oxopyrrolidine-5’-carboxylate.

Compound **4** was a yellow oil (MeOH) with a molecular formula of C_12_H_10_O_3_ according to its HR-ESI-MS (*m*/*z* 225.0648 [M + Na]^+^), implying eight degrees of unsaturation. ^1^H NMR (400 MHz) (Table 2; Appendix A) combined with HSQC revealed four aromatic protons at *δ*_H_ 7.72 (1H, d, *J* = 6.9 Hz, H-4), 7.77 (1H, td, *J* = 7.1, 1.0 Hz, H-5), 7.58 (1H, t, *J* = 7.6 Hz, H-6), and 7.81 (1H, d, *J* = 7.6 Hz, H-7); one hydroxy hydrogen at *δ*_H_ 5.28 (1H, d, *J* = 6.4 Hz, 8-OH); two oxymethine hydrogen at *δ*_H_ 5.67 (1H, d, *J* = 2.0 Hz, H-3) and 4.19 (1H, qd, *J* = 7.1, 2.0 Hz, H-8); one methine at *δ*_H_ 2.95 (1H, t, *J* = 2.8 Hz, H-11); and two methylene at *δ*_H_ 2.48 (2H, m, H-9). The ^13^C NMR (100 MHz) of DMSO-*d*_6_ revealed one ester carbonyl carbon signal at *δ*_C_ 170.4 (C-1); six sp^2^ hybrid carbons at *δ*_C_ 148.3 (C-3a), 134.4 (C-5), 129.5 (C-6), 126.7 (C-7a), 125.0 (C-7) and 123.3 (C-4); two oxygen-aliphatic signals at *δ*_C_ 82.4 (C-3) and 69.3 (C-8); two sp hybrid carbon signals at *δ*_C_ 81.6 and 73.6 (C-10 and C-11); and one saturated carbon at *δ*_C_ 24.2 (C-9).

The HMBC (Figure 2) correlations from H-6 to C-4/C-5/C-7a, from H-4 to C-3/C-6/C-7a, from H-5 to C-7/C-3a, from H-7 to C-1/C-5/C-3a, and from H-3 to C-1/C-3a/C-4/C-7a/C-8/C-9 established the 4a fragment. Correlations from 8-OH to C-3/C-8/C-9, from H-8 to C-3/C-3a/C-9/C-10, and from H-9 to C-3/C-8/C-11 implied the presence of a 4b fragment. Moreover, correlations between H-8 and C-3/C-3a and between H-3 and C-8/C-9 revealed the connecting position of segments 4a and 4b. The plane structure of compound **4** was elucidated.

The position of H-3 and H-8 of compound 5 was confirmed by the coupling constant of the compound. H-3 was *δ*_H_ 5.67 (1H, d, *J* = 2.0 Hz), and H-8 was *δ*_H_ 4.19 (1H, qd, *J* = 7.1, 2.0 Hz). Due to its coupling constant *J* = 2.0 Hz, the dominant conformation was obtained based on the carbon chain according to the Capon rule. The simulation confirmed that H-3 and H-8 of the compound were on the same side, which is a red configuration.

To clarify the absolute configuration of the C-3 position in compound **4**, CD spectroscopy was performed [34]. Its CD spectrum (Figure 4 and Appendix A) at 220 nm showed a positive Cotton curve, so the C-3 position was *S*-type. Moreover, H-3 and H-8 of compound **4** were red, so the C-8 position was the *S*-type. The absolute configuration of C-8 was further verified by the rhodium salt collation method, and there was a positive Cotton effect at 350 nm. Based on the above evidence, the C-8 position was *S*-type.

Compound **4** was elucidated and systemically named (3*S*,8*S*)-3-(1-hydroxy-but-3-ynyl)-3*H*-isobenzofuran-1-one.

Compound **5** was a yellow oil (MeOH) with a molecular formula of C_12_H_10_O_3,_ as the HR-ESI-MS (*m*/*z* 225.0648 [M + Na]^+^), with eight degrees of unsaturation. The ^1^H NMR and ^13^C NMR data (Table 2; Appendix A) indicated that compounds **4** and **5** have very similar structures. The only difference is the absolute configuration difference between them.

The position of H-3 and H-8 of compound 5 was confirmed by the coupling constant of the compound [34]. H-3 (*δ*_H_ 5.77) and H-8 (*δ*_H_ 3.91) showed the same coupling constant (*J* = 5.6 Hz). Due to its coupling constant *J* = 5.6 Hz, the dominant conformation was determined based on the carbon chain according to the Capon rule. The simulation confirmed that H-3 and H-8 of the compound were on the opposite side.

To clarify the absolute configuration of the C-3 position in compound **5**, CD spectroscopy was performed [34]. Its CD spectrum (Figure 4 and Appendix A) at 220 nm showed a positive Cotton effect, so we inferred that the C-3 position was *S*-type. Moreover, H-3 and H-8 of compound **5** were in the Threo form, so the C-8 position was inferred to be *R*. The absolute configuration of C-8 was further verified by the rhodium salt collation method, and there was a negative Cotton effect at 350 nm. Based on the above evidence, the C-8 position was ultimately determined to be the *R*-type.

Compound **5** was elucidated and systemically named (3*S*,8*R*)-3-(1-hydroxy-but-3-ynyl)-3*H*-isobenzofuran-1-one.

### 3.2. Antioxidant and TNF-α and NO Inhibitory Activities of the Components

In the DPPH assays, compounds **11** and **16** exhibited moderate activity, as shown in Table 3.

The LPS stimulation of NO production was also performed. Compounds **4**, **9**, **11** and **18** exhibited stronger inhibitory effects (IC_50_ < 35 μM), and indomethacin was the positive control (IC_50_ = 26.8 ± 1.3 μM). TNF-α is the main proinflammatory cytokine produced by macrophages and it has different proinflammatory effects on various cell types. In our research, the results showed that compounds **10** and **11** had better anti-inflammatory effects than the positive control group did. Compound **16** inhibits TNF-*α* production. The inhibitory activity of compound 16 was equivalent to that of the positive control, as shown in Table 4.

### 3.3. Chemical Composition Comparison

The compounds isolated from the rice fermentation of *Lentinus edodes* mycelium were compared with those obtained from the fruit body of Shiitake mushrooms. The secondary metabolites involved in the fermentation of *Lentinus edodes* mycelium are mainly isocoumarin compounds and their derivatives, which widely exist in nature but were isolated from *Lentinus edodes*. The chemical composition of Shiitake mushrooms was mainly composed of simple compounds with simple structural frameworks, and there were certain differences. In terms of activity, isocoumarin compounds have both anti-inflammatory and antioxidant activities. Moreover, mycelium produced more functional compounds with better activity.

## 4. Discussion

Shiitake mushrooms (*Lentinus edodes*) are rich in bioactive substances, which not only have high nutritional value, but also have medicinal properties. Therefore, this topic is of concern for researchers. At present, research on the active components of Shiitake mushrooms mainly focuses on the primary metabolites (polysaccharides, proteins, and polyunsaturated fatty acids) of *Lentinus edodes* mycelia and the nutritional components of the fruiting body itself [14]. We studied the fermentation metabolites of mushroom mycelia and the bioactive components of mushroom fruiting bodies and compared the compounds isolated from the two. In terms of the compound structure types, the metabolites produced during the fermentation of mushroom mycelia are mainly isocoumarin compounds and their derivatives. The chemical components isolated from the fruiting bodies of *Lentinus edodes* were mainly complexes with simple structural frameworks, followed by macrocyclic compounds. This indicates that there are differences in the main components of the two forms of shiitake mushrooms (fruiting bodies and mycelia).

Shiitake mushrooms have antioxidant, anti-aging, anti-inflammatory, and human immune functions. Shiitake mushrooms are widely used in food, health products, and medicine, and have broad research and development prospects [35,36,37,38,39,40]. In this study, the activity of the main compounds in the two forms of *Lentinus edodes* was tested, and the biological activity of the main compounds was analyzed and compared. In terms of compound activity, most of the isocoumarin compounds in mycelium metabolites have anti-inflammatory and antioxidant activities. In addition, the mycelia produced more functional compounds and exhibited better activity. This suggests that the metabolites of edible fungal mycelia are a new source of bioactive compounds.

In this study, new bioactive compounds of shiitake mushrooms with antioxidant and anti-inflammatory potential were identified, which were expected to provide a valuable theoretical reference for the rational development and utilization of edible and medicinal fungi represented by *Lentinus edodes*. In addition, we hope to provide new ideas for researchers to explore natural products with novel structures and excellent activities. However, it is undeniable that although the bioactive compounds in shiitake mushrooms have medicinal potential, whether there are side effects and whether they have medicinal properties still needs to be further explored by researchers.

In a word, shiitake mushrooms are a rich source of bioactive compounds with a high potential for development.

## 5. Conclusions

Our study systematically investigated the chemical composition of the Shiitake mushroom fruiting body and the solid-state fermentation of *Lentinus edodes* mycelium, and 5 new compounds and 15 known compounds were discovered. By analyzing the obtained compounds, we found that the non-polysaccharide components in the mycelium and the fruiting body of Shiitake mushrooms were different, and their activities were also different. Our results will provide a basis for the rational development of the functional constituents of the non-polysaccharides in Shiitake mushrooms and even the development of medicinal edible fungi.

## Figures and Tables

**Figure 1 jof-10-00552-f001:**
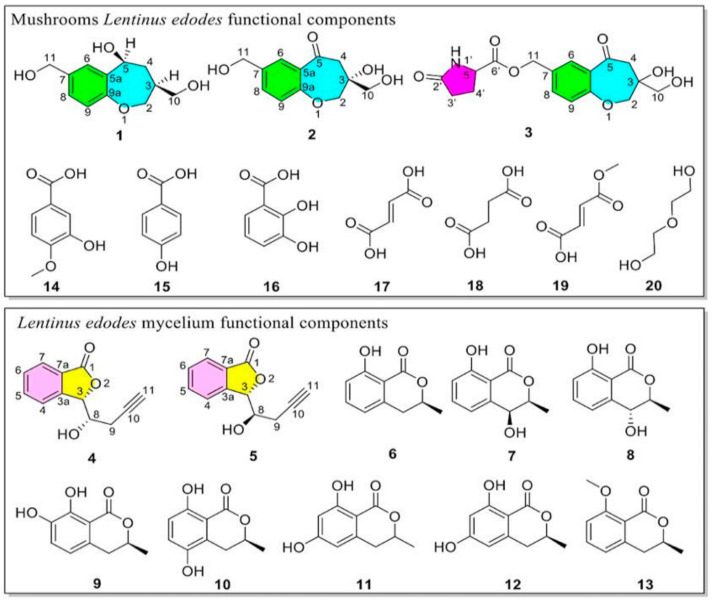
The structures of bioactive compounds **1–20**.

**Figure 2 jof-10-00552-f002:**
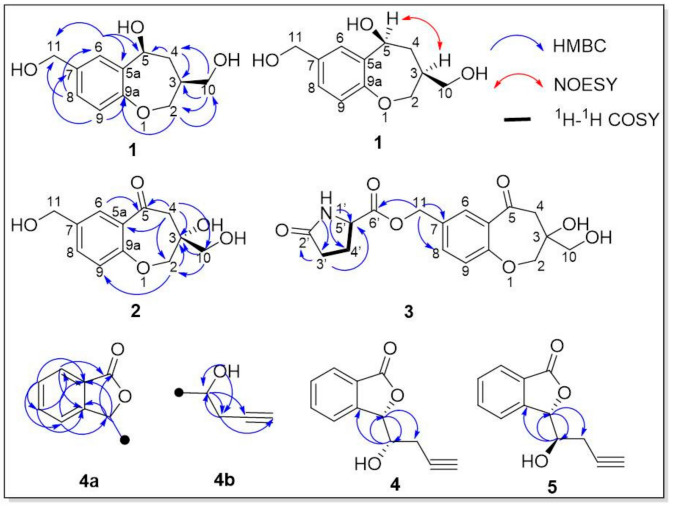
The key HMBC (blue) and NOESY (red) correlations of **1–5**.

**Figure 3 jof-10-00552-f003:**
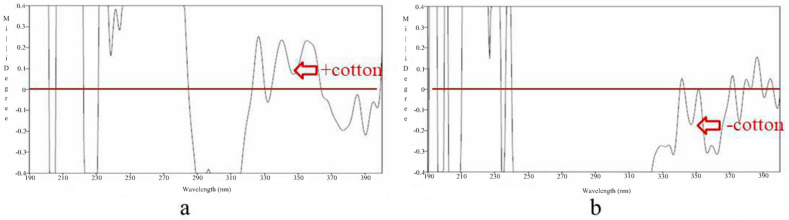
The CD spectrum of compounds **1–2** in CH_3_OH. (**a**) The CD spectrum of compound **1** after complexation with metal rhodium salt; the spectrum showed that there was a positive Cotton effect at 350 nm. (**b**) The CD spectrum of compound **2** after complexation with metal rhodium salt; the CD spectrum showed that there was a negative Cotton effect at 350 nm.

**Figure 4 jof-10-00552-f004:**
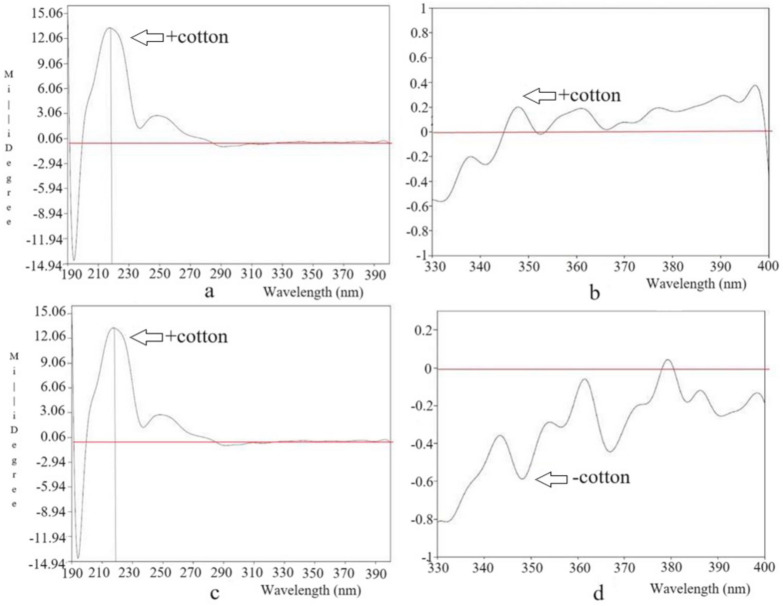
The CD spectrum of **4–5** in CH_3_OH. (**a**,**b**) The CD spectrum of compound **4** after complexation with metal rhodium salt; the spectrum showed that there was a positive Cotton effect at 350 nm. (**c**,**d**) The CD spectrum of compound **5** after complexation with metal rhodium salt; the spectrum showed that there was a negative Cotton effect at 350 nm.

**Table 1 jof-10-00552-t001:** The ^1^H-NMR (600 MHz) and ^13^C-NMR (150 MHz) spectrum data of compounds **1–3** in DMSO-*d*_6_.

NO	1		2		3	
	*δ*c	*δ*_H_ (m.; J in Hz)	*δ*c	*δ*_H_ (m.; J in Hz)	*δ*c	*δ*_H_ (m.; J in Hz)
1	-	-	-	-	-	-
2	75.0	3.17 (1H, d, 11.0) 4.31 (1H, dd, 11.0, 3.8)	80.7	4.17 (1H, d, 12.0) 3.84 (1H, d, 12.0)	80.1	4.19 (1H, m) 3.87 (1H, d, 12.0)
3	42.5	2.15 (1H, m)	76.7	-	76.7	-
4	39.6	2.02 (1H, d, 12.0) 1.28 (1H, q, 12.0)	52.5	2.92 (1H, d, 12.0) 2.76 (1H, d, 12.0)	52.5	2.93 (1H, d, 12.0) 2.77 (1H, d, 12.0)
5	68.7	4.72 (1H, d, 11.0)	196.6	-	196.3	-
5a	138.7	-	127.6	-	127.9	-
6	124.7	7.48 (1H, s)	127.3	7.62 (1H, d, 2.3)	129.6	7.68 (1H, d, 2.3)
7	137.6	-	137.3	-	130.5	-
8	125.9	7.06 (1H, dd, 8.5, 2.0)	132.7	7.40 (1H, dd, 8.5, 2.3)	134.4	7.50 (1H, dd, 8.3, 2.3)
9	120.5	6.85 (1H, d, 8.1)	120.7	7.05 (1H, d, 8.5)	121.2	7.12 (1H, d, 8.3)
9a	156.5	-	162.3	-	163.2	-
10	62.8	3.31 (1H, m) 3.23 (1H, m)	67.1	3.37 (2H, m)	67.0	3.40 (2H, m)
11	63.4	4.43 (2H, s)	62.5	4.44 (2H, s)	66.0	5.11 (2H, d, 12.0)
2′					177.4	-
3′					29.3	2.10 (2H, m)
4′					25.0	2.33 (1H, m) 1.95 (1H, m)
5′					55.2	4.20 (1H, m)
6′					173.2	-

**Table 2 jof-10-00552-t002:** The ^1^H-NMR (400 MHz) and ^13^C-NMR (100 MHz) spectrum data of compounds **4–5** in DMSO-*d*_6_.

	4		5	
NO	*δ* _C_	*δ*_H_ (m.; *J* in Hz)	*δ* _C_	*δ*_H_ (m.; *J* in Hz)
1	170.4	-	170.1	-
2	-	-	-	-
3	82.4	5.67 (1H, d, 2.0)	82.4	5.77 (1H, d, 5.6)
4	123.3	7.72 (1H, d, 6.9)	124.3	7.73 (1H, d, 6.9)
5	134.4	7.77(1H, td, 7.1, 1.0)	134.4	7.77 (1H, td, 7.1, 1.0)
6	129.5	7.58 (1H, t, 7.6)	129.7	7.61 (1H, t, 7.6)
7	125.0	7.81 (1H, d, 7.6)	125.2	7.83 (1H, d, 7.6)
3a	148.3	-	147.6	-
7a	126.6	-	126.4	-
8	69.3	4.19 (1H, qd, 7.1, 2.0)	70.1	3.91 (1H, m, 11.6, 5.6)
9	24.2	2.48 (2H, m)	23.8	2.48 (2H, m)
10	81.6	-	81.3	-
11	73.6	2.95 (1H, t, 2.8)	73.6	2.89 (1H, t, 2.8)
8-OH	-	5.28 (1H, d, 6.4)	-	5.53 (1H, d, 5.6)

**Table 3 jof-10-00552-t003:** Determination of DPPH antioxidant activity (IC_50_).

Compounds	DPPH (IC_50_, μg/mL)
**4**	54.3 ± 1.8
**7**	78.9 ± 1.7
**8**	69.5 ± 0.9
**9**	54.3 ± 1.8
**10**	48.6 ± 1.2
**11**	25.2 ± 0.5
**12**	54.3 ± 1.8
**14**	52.8 ± 1.4
**15**	48.4 ± 2.1
**16**	34.7 ± 2.5
**17**	46.2 ± 0.9
**18**	>100
**19**	>100
**20**	98.7 ± 3.6
Trolox	10.8 ± 0.5

**Table 4 jof-10-00552-t004:** Inhibition on NO production of isolated compounds from Shiitake mushrooms (*Lentinus edodes*).

Compounds	NO Inhibitory Assay (IC_50_) μM	TNF-*α* Inhibitory Assay (IC_50_) μM
**4**	34.7 ± 2.5	>100
**7**	48.6 ± 3.4 *	>100
**8**	52.1 ± 3.6 *	>100
**9**	30.2 ± 2.5	94.5 ± 1.6
**10**	35.8 ± 1.7	75.3 ± 2.7 *
**11**	28.1 ± 2.2 *	68.1 ± 1.8 *
**12**	58.4 ± 1.9 *	>100
**14**	83.1 ± 3.1 **	98.4 ± 1.3
**15**	79.8 ± 2.8 **	-
**16**	45.7 ± 1.2 *	89.3 ± 1.8
**17**	>100	>100
**18**	29.9 ± 1.2	>100
**19**	89.6 ± 1.7 **	-
**20**	91.8 ± 2.3 **	-
Indomethacin	26.8 ± 1.3	88.5 ± 2.1

Significantly different compared to the indomethacin * *p* < 0.05, ** *p* < 0.01.

## Data Availability

The original data presented in the study are included in the article; further inquiries can be directed to the corresponding author.

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
