# Peer review of "Chemical Composition, Antioxidant and Anti-Inflammatory Activity of Shiitake Mushrooms (Lentinus edodes)"

_jof, 2024, doi:10.3390/jof10080552_

Round 1
Reviewer 1 Report
Re: Chemical composition and function evaluation of mycelium and 2 fruiting body of Lentinula edodes. Journal of Foods
The study evaluated the phytochemical profile, antioxidant and anti-inflammatory properties of Shiitake Mushroom (Lentinus edodes). Some new compounds and several known compounds were isolated using various spectroscopic methods such as HPLC, NMR and NOESY. The classes of the compounds isolated includes isocoumarins, macrocycles, polyphenols and fatty acids. The antioxidant activity was evaluated against a stable DPPH free radical while the anti-inflammatory activity was evaluated in vitro against both NO and TNF-α. Compounds 10, 11, and 16 exhibited potent TNF-α inhibitory effects, and com-25 pounds 4, 9, 11 and 18 exhibited strong NO inhibitory effects.
Although the manuscript is of high quality (novel) and adds value to the Scientific body of knowledge, it must not be accepted in its current form. I recommend major revisions. I have the following comments to add value to the manuscript.
1. Tittle. The tittle is not well presenting the actual work done and not catchy at all. I propose the title to read “Chemical composition, antioxidant and anti-inflammatory activity of Shiitake Mushroom (Lentinus edodes).”.
2. Abstract: The abstract is not well written hence not catchy and not reflection of the work done. Please rewrite and where possible, mention the biologically active compounds rather than the classes of compounds. Lentinus edodes appears in line 17, 18, 20 and 26, both as a repetition and not in italics.
3. Introduction: It doesn’t necessarily introduce the Mushroom used. Furthermore, the name Lentinus edodes is repeated a lot more times without a reason. Please see line 37, 38, 40, 41, 42, 43, 45 and 47. It makes the article very untidy and less Scientific. I believe Figure 1 is part of the results and can never be presented in the introductory part of the manuscript.
4. Materials and methods: It is not systematic. Please rewrite this part. Afterall, why purchase? Line 77-79?
5. Results. Line 257. Relative to what?
6. Amazingly, the results found in the current study were not compared to any data from the Literature.
Re: Chemical composition and function evaluation of mycelium and 2 fruiting body of Lentinula edodes. Journal of Foods
The study evaluated the phytochemical profile, antioxidant and anti-inflammatory properties of Shiitake Mushroom (Lentinus edodes). Some new compounds and several known compounds were isolated using various spectroscopic methods such as HPLC, NMR and NOESY. The classes of the compounds isolated includes isocoumarins, macrocycles, polyphenols and fatty acids. The antioxidant activity was evaluated against a stable DPPH free radical while the anti-inflammatory activity was evaluated in vitro against both NO and TNF-α. Compounds 10, 11, and 16 exhibited potent TNF-α inhibitory effects, and com-25 pounds 4, 9, 11 and 18 exhibited strong NO inhibitory effects.
Although the manuscript is of high quality (novel) and adds value to the Scientific body of knowledge, it must not be accepted in its current form. I recommend major revisions. I have the following comments to add value to the manuscript.
1. Tittle. The tittle is not well presenting the actual work done and not catchy at all. I propose the title to read “Chemical composition, antioxidant and anti-inflammatory activity of Shiitake Mushroom (Lentinus edodes).”.
2. Abstract: The abstract is not well written hence not catchy and not reflection of the work done. Please rewrite and where possible, mention the biologically active compounds rather than the classes of compounds. Lentinus edodes appears in line 17, 18, 20 and 26, both as a repetition and not in italics.
3. Introduction: It doesn’t necessarily introduce the Mushroom used. Furthermore, the name Lentinus edodes is repeated a lot more times without a reason. Please see line 37, 38, 40, 41, 42, 43, 45 and 47. It makes the article very untidy and less Scientific. I believe Figure 1 is part of the results and can never be presented in the introductory part of the manuscript.
4. Materials and methods: It is not systematic. Please rewrite this part. Afterall, why purchase? Line 77-79?
5. Results. Line 257. Relative to what?
6. Amazingly, the results found in the current study were not compared to any data from the Literature.
Author Response
Point-by-point response
We gratefully thank the editor and reviewers who took the time to make constructive comments and useful suggestions, which greatly improved the quality of the manuscript and enabled us to improve the manuscript. Every revision suggestion and comment made by the reviewer was accurately incorporated and considered. Below the reviewer's comments were point-by-point responses, marked with changes. What’s more, we have also reviewed and polished the language of the manuscript throughout, and these changes will not affect the content and framework of the paper. Here we do not list the changes, but in the revised paper marked in red.
Reviewer 1
The study evaluated the phytochemical profile, antioxidant and anti-inflammatory properties of Shiitake Mushroom (Lentinus edodes). Some new compounds and several known compounds were isolated using various spectroscopic methods such as HPLC, NMR and NOESY. The classes of the compounds isolated includes isocoumarins, macrocycles, polyphenols and fatty acids. The antioxidant activity was evaluated against a stable DPPH free radical while the anti-inflammatory activity was evaluated in vitro against both NO and TNF-α. Compounds 10, 11, and 16 exhibited potent TNF-α inhibitory effects, and compounds 4, 9, 11 and 18 exhibited strong NO inhibitory effects.
Although the manuscript is of high quality (novel) and adds value to the Scientific body of knowledge, it must not be accepted in its current form. I recommend major revisions. I have the following comments to add value to the manuscript.
Answer: First of all, thank you for your recognition of our manuscript. We greatly appreciate your constructive comments and useful suggestions, which greatly improves the quality of the manuscript. Each suggested revision and comments, brought forward by you was accurately incorporated and considered. All changes in the manuscript are marked using red font.
Question 1: Tittle. The tittle is not well presenting the actual work done and not catchy at all. I propose the title to read “Chemical composition, antioxidant and anti-inflammatory activity of Shiitake Mushroom (Lentinus edodes).”.
Answer: Thank you for your positive comments and valuable suggestions to improve the quality of our manuscript. We have revised it in the manuscript.
Question 2: Abstract: The abstract is not well written hence not catchy and not reflection of the work done. Please rewrite and where possible, mention the biologically active compounds rather than the classes of compounds. Lentinus edodes appears in line 17, 18, 20 and 26, both as a repetition and not in italics.
Answer: We sincerely thank you for your useful comments. We have re-written this part according to your suggestion. And thanks for your careful checks. We are sorry for our carelessness. Based on your comments, we have made the corrections.
Question 3: Introduction: It doesn’t necessarily introduce the Mushroom used. Furthermore, the name Lentinus edodes is repeated a lot more times without a reason. Please see line 37, 38, 40, 41, 42, 43, 45 and 47. It makes the article very untidy and less Scientific. I believe Figure 1 is part of the results and can never be presented in the introductory part of the manuscript.
Answer: According to your professional comments, we have made extensive modifications of this part in our manuscript. Thank you again for your positive comments and valuable suggestions to improve the quality of our manuscript.
Question 4: Materials and methods: It is not systematic. Please rewrite this part. Afterall, why purchase? Line 77-79?
Answer: We greatly appreciate your constructive comments, which greatly improves the quality of the manuscript. We have re-written this part according to your suggestion.
Question 5: Results. Line 257. Relative to what?
Answer: We are very sorry for our negligence of this part. Indomethacin was positive control (IC50 = 26.8 ± 1.3 μM), compared with indomethacin, compounds 4, 9, 11 and 18 exhibited stronger NO inhibitory activity (IC50 < 35 μM).
Question 6: Amazingly, the results found in the current study were not compared to any data from the Literature.
Answer: We are sincerely apologized for our negligence of this part. We have supplemented the missing literature.
[1] Frelek J; Szczepek W.J. [Rh2(OCOCF3)4] as an auxiliary chromophore in chiroptical studies on steroidal alcohols. Tetrahe-dron Asymmetry 1999, 10, 1507-1520. https://doi.org/10.1016/S0957-4166(99)00115-9.
[2] Frelek J.; Jagodziński J.; Meyer-Figge H. Chiroptical properties of binuclear rhodium com-plexes of lanostane alcohols. Chirality 2001, 13, 313-321. https://doi.org/10.1002/chir.1037.
[3] Ding G.; Liu S.; Guo L. Antifungal metabolites from the plant endophytic fungus Pestalotiopsis foedan. J Nat Prod 2008, 71, 615-618. https://doi.org/10.1021/np070590f.
Finally, we tried our best to improve the manuscript and made some changes marked in red in revised paper which will not influence the content and framework of the paper. We appreciate for Reviewers’ warm work earnestly, and hope the correction will meet with approval. Once again, thank you very much for your comments and suggestions.
Reviewer 2 Report
This work deals with the chemical composition and function evaluation of mycelium and fruiting body of Lentinula edodes. Sophisticated analytical instruments were used, including LC-MS, H NMR and C NMR, along with the determination of critical biomarkers, such as antioxidant activity potential, tumor necrosis factor and cellular metabolic activity. However, the Materials and Methods section lacks specific information about the reagents used as well as the instrumentation and operation conditions of LC-MS and NMR techniques. Also, a substantial section of an article, the Discussion section, is completely missing from the manuscript. It should be added either as a separate section (Discussion) or combined with the results (i.e. Results and Discussion), having a lot of critical discussion of the findings, illustrating the challenges and comments on future directions, along with a significant number of References.
Specific comments that need to be addressed to improve the quality of the manuscript are given below:
-I suggest you improve the title by writing to the end "...of the edible fungus Lentinula edodes" or "...of the edible mushroom Lentinula edodes" or something like that...
-Write in italics the "Lentinus edodes" wherever mentioned in the Abstract.
Lines 19-20: Continue the phrase "....were isolated" by adding also "and detected by..." and give the detection systems used to determine the compounds (e.g. HPLC, LC-MS, H NMR, C NMR)
-Lines 20-22: Please remove the numbers in the parenthesis. They are not necessary in the Abstract section and confuse the reader.
-Lines 32-33: Repetition of the phrase "commonly used" within one sentence. Please rewrite the sentence accordingly to avoid duplication.
-Line 34: The term "functional physiological regulatory food" is strange. Please rewrite the sentence using formal terms.
-Lines 47-50: Better explain what the mycelium and fruiting body are. Provide clear definitions.
-Lines 55-57: Please rewrite the sentence. It is not clearly written and contains expression mistakes.
-Figure 1: Even if the image was created by you, you must cite the reference from which you have found the information about the chemical structures.
-Lines 63-71: Please give a separate sub-section about the Reagents. Provide details about the name of each reagent, the exact % purity of each reagent as well as the name of the company, city etc.. The phrase "The HPLC reagents are all chromatographic pure (100% purity), while the other reagents are analytical pure" in Lines 68-69 is very simplified and informal.
-Lines 63-141: You are providing data for LC-MS, H NMR and C NMR. Where is the information about the instruments and the analytical conditions for all these, such as instrument model, detectors, companies, columns, ESI source, ionization mode, flow rates etc? Only the information about HPLC was merely mentioned at the beginning of the section. You should also include the HPLC operation conditions, i.e. flow rate, temperature, specific wavelength used to obtain the chromatograms etc.
-Line 81: What do you mean 170 r? Please give the velocity value of the shaker in rpm and the name of company, city etc.
-Lines 81-82: Please give the exact inoculum size, i.e. cells/mL or spores/mL, and the volume added of this inoculum to the 5 mL media. Also, mention how did you measure the inoculum size.
-Line 83: Why so long incubation period (45 days)? Can you explain? Did you measure any parameters in order to choose the number of days? Please specify.
-Lines 85-86: Please give the exact conditions of the extraction, i.e. volumes, use of vortex and/or centrifuge (along with min, rpm, temperature) etc
-Figures 2, 3 and 4. Improve the texts of the Figure legends. For example, "spectrum of 1-2" is not a formal comment. These are the "compounds" 1 and 2 but it is not clearly written. Also, transfer the Figure 3 legend closer to the Figure 3.
-A Discussion section, following the Results section, is completely absent from the manuscript. This work lacks critical critical discussion based on comparison with literature data. A significant number of literature data should be used to compare and explain the findings with criticality. According to the Journal's Guide for Authors, this missing section is mandatory. Please add a large Discussion section (of similar text length to the Results) that sufficiently evaluates the findings of your research, illustrates the challenges and comments on future considerations.
Author Response
Point-by-point response
We gratefully thank the editor and reviewers who took the time to make constructive comments and useful suggestions, which greatly improved the quality of the manuscript and enabled us to improve the manuscript. Every revision suggestion and comment made by the reviewer was accurately incorporated and considered. Below the reviewer's comments were point-by-point responses, marked with changes. What’s more, we have also reviewed and polished the language of the manuscript throughout, and these changes will not affect the content and framework of the paper. Here we do not list the changes, but in the revised paper marked in red.
Reviewer 2
Question 1: I suggest you improve the title by writing to the end "...of the edible fungus Lentinula edodes" or "...of the edible mushroom Lentinula edodes" or something like that...
Answer: Thank you for your valuable suggestions to improve the quality of our manuscript. Another reviewer also made valuable suggestions on this, according to the suggestions of both reviewers, we have changed the title to “Chemical composition, antioxidant and anti-inflammatory activity of Shiitake Mushroom (Lentinus edodes)”.
Question 2: Write in italics the "Lentinus edodes" wherever mentioned in the Abstract.
Answer: Thanks for your careful checks. We are sorry for our carelessness. Based on your comments, we have made the corrections.
Question 3: Lines 19-20: Continue the phrase ".... were isolated" by adding also "and detected by..." and give the detection systems used to determine the compounds (e.g. HPLC, LC-MS, H NMR, C NMR)
Answer: We greatly appreciate your constructive comments and useful suggestions, which greatly improves the quality of the manuscript. We have re-written Abstract part.
Question 4: -Lines 20-22: Please remove the numbers in the parenthesis. They are not necessary in the Abstract section and confuse the reader.
Answer: Thank you for your professional suggestions. The numbers in brackets represent the number of the compound, which will be more convenient to use in the subsequent description of the activity experiment. We are sincerely apologized that the previous Abstract part confused you, and we have re-written Abstract part.
Question 5: -Lines 32-33: Repetition of the phrase "commonly used" within one sentence. Please rewrite the sentence accordingly to avoid duplication.
Answer: We greatly appreciate your constructive comments and useful suggestions, which greatly improves the quality of the manuscript. We have re-written Introduction part.
Question 6: -Line 34: The term "functional physiological regulatory food" is strange. Please rewrite the sentence using formal terms.
Answer: We greatly appreciate your constructive comments and useful suggestions, which greatly improves the quality of the manuscript. We have re-written Introduction part.
Question 7: -Lines 47-50: Better explain what the mycelium and fruiting body are. Provide clear definitions.
Answer: The growth of Lentinus edodes can be divided into two stages: the vegetative phase (mycelium or mycelial growth) and reproductive phase (fruiting bodies). After the scattered spores invaded the substrate, the hyphae, which were only visible under a microscope, continually grew and branched to form mycelia, and the fruiting body grew out of the subterranean mycelium through a process called fructification. The formed fruiting bodies (bottom of the cap) are sporulation structures that are edible parts of edible fungi.
[1] Sánchez C. Reactive oxygen species and antioxidant properties from mushrooms. Synth Syst Biotechnol 2016, 24, 13-22. https://doi.org/10.1016/j.synbio.2016.12.001.
[2] Berger, R.G.; Bordewick, S.; Krahe, N.-K.; Ersoy, F. Mycelium vs. Fruiting Bodies of Edible Fungi-A Comparison of Metabolites. Microorganisms 2022, 10, 1379. https://doi.org/10.3390/microorganisms10071379.
Question 8: -Lines 55-57: Please rewrite the sentence. It is not clearly written and contains expression mistakes.
Answer: We greatly appreciate your constructive comments and useful suggestions, which greatly improves the quality of the manuscript. We have re-written Introduction part.
Question 9: -Figure 1: Even if the image was created by you, you must cite the reference from which you have found the information about the chemical structures.
Answer: Thank you for your valuable suggestions. We have revised it in the manuscript, and we adjusted the position of the appearance of Figure 1 to the Results part, and supplemented the related literature of compounds in it.
[1] Reveglia, P.; Masi, M.; Evidente, A. Melleins—Intriguing Natural Compounds. Biomolecules, 2020, 10, 772. https://doi.org/10.3390/biom10050772.
[2] Naman D., Mireille F., Laurent D., Basil D., Sameer A.S. M. In silico studies on Epicoccum spp. Secondary metabolites as potential drugs for mucormycosis, Results in Chemistry, 2024, 7, 101420, https://doi.org/10.1016/j.rechem.2024.101420.
[3] Lazić, V.; Klaus, A.; Kozarski, M.; Doroški, A.; Tosti, T.; Simić, S.; Vunduk, J. The Effect of Green Extraction Technologies on the Chemical Composition of Medicinal Chaga Mushroom Extracts. J. Fungi, 2024, 10, 225. https://doi.org/10.3390/jof10030225
[4] Baptista, F.; Campos, J.; Costa-Silva, V.; Pinto, A.R.; Saavedra, M.J.; Ferreira, L.M.; Rodrigues, M.; Barros, A.N. Nutraceutical Potential of Lentinula edodes’Spent Mushroom Substrate: A Comprehensive Study on Phenolic Composition, Antioxidant Activity, and Antibacterial Effects. J. Fungi, 2023, 9, 1200. https://doi.org/10.3390/jof9121200
[5] Wu, F.; Wang, H.; Chen, Q.; Pang, X.; Jing, H.; Yin, L.; Zhang, X. Lignin Promotes Mycelial Growth and Accumulation of Polyphenols and Ergosterol in Lentinula edodes. J. Fungi, 2023, 9, 237. https://doi.org/10.3390/jof9020237
[6] Ma, Z.-L.; Yu, Z.-P.; Zheng, Y.-Y.; Han, N.; Zhang, Y.-H.; Song, S.-Y.; Mao, J.-Q.; Li, J.-J.; Yao, G.-S.; Wang, C.-Y. Bioactive Alpha-Pyrone and Phenolic Glucosides from the Marine-Derived Metarhizium sp. P2100. J. Fungi, 2023, 9, 28. https://doi.org/10.3390/jof9010028
Question 10: -Lines 63-71: Please give a separate sub-section about the Reagents. Provide details about the name of each reagent, the exact % purity of each reagent as well as the name of the company, city etc. The phrase "The HPLC reagents are all chromatographic pure (100% purity), while the other reagents are analytical pure" in Lines 68-69 is very simplified and informal.
Answer: We greatly appreciate your constructive comments, which greatly improves the quality of the manuscript. We have re-written this part according to your suggestion.
Question 11: -Lines 63-141: You are providing data for LC-MS, H NMR and C NMR. Where is the information about the instruments and the analytical conditions for all these, such as instrument model, detectors, companies, columns, ESI source, ionization mode, flow rates etc? Only the information about HPLC was merely mentioned at the beginning of the section. You should also include the HPLC operation conditions, i.e. flow rate, temperature, specific wavelength used to obtain the chromatograms etc.
Answer: We greatly appreciate your constructive comments, which greatly improves the quality of the manuscript. Considering your suggestion, we have re-written this part.
Question 12: -Line 81: What do you mean 170 r? Please give the velocity value of the shaker in rpm and the name of company, city etc.
Answer: We were really sorry for our careless mistakes. We have corrected the “170 r” into “170 rpm”. In this study, we used a constant temperature oscillation chamber. We have added the manufacturer's name of the constant temperature oscillator chamber. Thank you for your reminder.
Question 13: -Lines 81-82: Please give the exact inoculum size, i.e. cells/mL or spores/mL, and the volume added of this inoculum to the 5 mL media. Also, mention how did you measure the inoculum size.
Answer: We are apologized for the confusion caused by the strain fermentation part, and now we will answer this question. Generally, the seed liquid was added according to the proportion of 5%-10% of the fermentation liquid. The fermentation liquid in each bottle of culture medium was 100 mL, so 5-10 mL of seed liquid need to be inoculated in each bottle of culture medium. The relevant literature also showed that under the same fermentation conditions, 5 mL seed liquid inoculated in each bottle was more appropriate.
[1] Xu, X.; Dong, Y.; Yang, J.; Wang, L.; Ma, L.; Song, F.; Ma, X. Secondary Metabolites from Marine-Derived Fungus Penicillium rubens BTBU20213035. J. Fungi 2024, 10, 424.
[2] Jiyoon P., Jiseong K., Sunghoon H., Daehyun O., Young E., Sang-Jip N., Hyeung-geun P., Min J. L., and Dong-Chan O. Sadopeptins A and B, Sulfoxide- and Piperidone-Containing Cyclic Heptapeptides with Proteasome Inhibitory Activity from a Streptomyces sp. J. Nat. Prod. 2023, 86, 612-620.
Question 14: -Line 83: Why so long incubation period (45 days)? Can you explain? Did you measure any parameters in order to choose the number of days? Please specify.
Answer: We are apologized for the confusion caused by the strain fermentation part, and now we will answer this question. The fermentation time of fungi in rice culture medium is generally 30-45 days, and the specific time depends on the fermentation species. In practical application, according to different types of fungi and experimental needs, it may be necessary to adjust the fermentation days to achieve the best effect. In the process of fermentation, we had observed the growth and metabolism of the strain every day. After 45 days of fermentation, the strain metabolized the medium completely.
[1] Abulaizi, A.; Wang, R.; Xiong, Z.; Zhang, S.; Li, Y.; Ge, H.; Guo, Z. Secondary Metabolites with Agricultural Antagonistic Potential from Aspergillus sp. ITBBc1, a Coral-Associated Marine Fungus. Mar. Drugs 2024, 22, 270.
Question 15: -Lines 85-86: Please give the exact conditions of the extraction, i.e. volumes, use of vortex and/or centrifuge (along with min, rpm, temperature) etc
Answer: We greatly appreciate your constructive comments, which greatly improves the quality of the manuscript. Considering your suggestion, we have re-written this part.
Question 16: -Figures 2, 3 and 4. Improve the texts of the Figure legends. For example, "spectrum of 1-2" is not a formal comment. These are the "compounds" 1 and 2 but it is not clearly written. Also, transfer the Figure 3 legend closer to the Figure 3.
Answer: Thank you for your professional suggestions. We have revised it in the manuscript.
Question 17: -A Discussion section, following the Results section, is completely absent from the manuscript. This work lacks critical critical discussion based on comparison with literature data. A significant number of literature data should be used to compare and explain the findings with criticality. According to the Journal's Guide for Authors, this missing section is mandatory. Please add a large Discussion section (of similar text length to the Results) that sufficiently evaluates the findings of your research, illustrates the challenges and comments on future considerations.
Answer: Thank you for your professional suggestions. We have revised it in the manuscript.
- Discussion
Shiitake mushrooms (Lentinus edodes) are rich in bioactive substances, which not only have high nutritional value, but also have medicinal properties. Therefore, it has been a concern for researchers. At present, research on the active components of Shiitake mushrooms mainly focuses on the primary metabolites (polysaccharides, proteins, and polyunsaturated fatty acids) of Lentinus edodes mycelia and the nutritional components of the fruiting body itself [35]. Shiitake mushrooms are widely used in food, health products, and medicine and have broad research and development prospects [36-41]. In this study, new bioactive compounds of shiitake mushrooms with antioxidant and anti-inflammatory potential were identified, which were expected to provide a valuable theoretical reference for the rational development and utilization of edible and medicinal fungi represented by Lentinus edodes. However, it is undeniable that although the bioactive compounds in shiitake mushrooms have medicinal potential, whether there are side effects and whether they have medicinal properties still needs to be further explored by researchers. In a word, shiitake mushrooms are a rich source of bioactive compounds with a high potential for development.
Finally, we tried our best to improve the manuscript and made some changes marked in red in revised paper which will not influence the content and framework of the paper. We appreciate for Reviewers’ warm work earnestly, and hope the correction will meet with approval. Once again, thank you very much for your comments and suggestions.
Round 2
Reviewer 1 Report
The manuscript is well revised and ready for publication.
The proposed amendments have been effective. The paper is ready for publication.
Author Response
First of all, thank you for your recognition of our manuscript. We greatly appreciate your constructive comments and useful suggestions, which greatly improve the quality of the manuscript. We would like to express our sincere thanks to the reviewers for their enthusiastic work. Finally, thank you again for your comments and suggestions.
Reviewer 2 Report
The content of the manuscript has been significantly improved and can be further considered for publication.
My specific comments for the revised manuscript are given below.
-Lines 22-25: Please write either the word compounds before the numbers or give the exact name of the identified compounds. Just the number without a proper mention does not provide clarity.
-Lines 97-98 and 106-108: Mention that this was a "liquid-liquid extraction" process, the minutes of mixing, if a vortex or separating funnel was used for the mixing, and if centrifugation was used to separate the organic layer from the solid material.
-Lines 315-329: The Discussion section should be transferred after the Results and before the Conclusion. Add 2 more paragraphs to the discussion section. Just 1 paragraph is too short. Explain in more detail the phenomena detected in the Results and the information regarding the importance of the identified compounds.
Author Response
Reviewer 2
Question 1: Lines 22-25: Please write either the word compounds before the numbers or give the exact name of the identified compounds. Just the number without a proper mention does not provide clarity.
Answer: We greatly appreciate your constructive comments and useful suggestions, which greatly improves the quality of the manuscript. We have revised it in the manuscript.
Question 2: Lines 97-98 and 106-108: Mention that this was a "liquid-liquid extraction" process, the minutes of mixing, if a vortex or separating funnel was used for the mixing, and if centrifugation was used to separate the organic layer from the solid material.
Answer: We are apologized for the confusion caused by the extraction part, and now we will answer this question. This was the process of solid-liquid extraction. The organic reagent ethyl acetate was used to extract the secondary metabolites from the mycelium of Lentinus edodes and the bioactive compounds from the fruiting body of shiitake mushrooms. The extraction rate was increased by stirring with glass rods and ultrasonic-assisted methods, and the operation was repeated for 3 times to make the extraction more thorough. The organic extract was separated from the solid material by the way of cotton cloth filtration, and the crude extract was obtained by concentrating the extract. The crude extract was studied later.
Question 3: Lines 315-329: The Discussion section should be transferred after the Results and before the Conclusion. Add 2 more paragraphs to the discussion section. Just 1 paragraph is too short. Explain in more detail the phenomena detected in the Results and the information regarding the importance of the identified compounds.
Answer: We greatly appreciate your professional comments, which greatly improves the quality of the manuscript. Considering your suggestion, we have perfected the manuscript.
Finally, we tried our best to improve the manuscript and made some changes marked in red in revised paper which will not influence the content and framework of the paper. We appreciate for Reviewers’ warm work earnestly, and hope the correction will meet with approval. Once again, thank you very much for your comments and suggestions.
